# Biomass Allocation into Woody Parts and Foliage in Young Common Aspen (*Populus tremula* L.)—Trees and a Stand-Level Study in the Western Carpathians

**Bohdan Konôpka** [1,2]**, Jozef Pajtík** [1]**, Vladimír Šebeň** [1,]*[ID]**, Peter Surový** [2][ID] **and Katarína Merganičová** [2,3]

[1] Forest Research Institute, National Forest Centre, T.G. Masaryka 22, SK-960 01 Zvolen, Slovakia; bohdan.konopka@nlcsk.org (B.K.); jozef.pajtik@nlcsk.org (J.P.)
[2] Faculty of Forestry and Wood Sciences, Czech University of Life Sciences Prague, Kamýcká 129, CZ-165 000 Prague, Czech Republic; psurovy@gmail.com (P.S.); merganicova@tuzvo.sk (K.M.)
[3] Forestry Faculty, Technical University Zvolen, T.G. Masaryka 24, SK-960 01 Zvolen, Slovakia
* Correspondence: seben@nlcsk.org

**Abstract:** Our research of common aspen (*Populus tremula* L.) focused on the forested mountainous area in central Slovakia. Forest stands (specifically 27 plots from 9 sites) with ages between 2 and 15 years were included in measurements and sampling. Whole tree biomass of aspen individuals was destructively sampled, separated into tree components (leaves, branches, stem, and roots), and then dried and weighed. Subsamples of fresh leaves from three crown parts (upper, middle, and lower) were scanned, dried, and weighed. Allometric biomass models with stem base diameter as an independent variable were derived for individual tree components. Basic foliage traits, i.e., leaf mass, leaf area, and specific leaf area, were modelled with regard to tree size and leaf position within the crown. Moreover, biomass stock of the woody parts and foliage as well as the leaf area index were modelled using mean stand diameter as an independent variable. Foliage traits changed with both tree size and crown part. Biomass models showed that foliage contribution to total tree biomass decreased with tree size. The total foliage area of a tree increased with tree size, reaching its maximum value of about 12 m$^2$ for a tree with a diameter of 120 mm. Leaf area index increased with mean stand diameter, reaching a maximum value of 13.5 m$^2$ m$^{-2}$. Since no data for biomass allocation for common aspen had been available at either the tree or stand levels, our findings might serve for both theoretical (e.g., modelling of growth processes) and practical (forestry and agro-forestry stakeholders) purposes.

**Keywords:** tree components; biomass models; stem base diameter; specific leaf area; leaf area index

## 1. Introduction

Knowledge on the patterns of biomass partitioning of a variety of species is of high importance for carbon reporting [1], tree physiology, plant ecology, and process-based modelling [2], and has also applications for forestry management [3]. Although generalised biomass models can be used to estimate biomass stock in forests stands [4], locally fitted models are recommended by the IPCC to minimise the bias of estimations [5]. In addition, young stands are characterised by fast changes in biomass allocation in individual tree compartments, due to which biomass equations developed for older stands are not appropriate [6–8].

Considering a range of forest tree species, a lot of works focused on biomass allocation and foliage traits with regard to a number of different factors, e.g., competition in a stand [9,10]. Few authors [11–13] used the ratio between foliage dry mass and total dry plant biomass (leaf mass ratio; LMR) or between foliage area and total plant dry biomass (leaf area ratio; LAR) to link them to ecological

and production processes. Similarly, specific leaf area (ratio between leaf area and dry leaf mass; SLA) has often been studied in plant ecology as one of the adaptation strategy indicators [14]. Canopy leaf area of forests serves as a dominant physical control over primary production, transpiration, energy exchange, and other physiological attributes related to a variety of ecosystem processes; hence, it is a substantial element of ecological studies [15]. Here, the leaf area index (LAI, defined as the amount of leaf area in the canopy per unit ground area) is the main parameter of the canopy [16]. None of these indicators has been studied in common aspen either at the tree or the stand level.

Common aspen, or also named Eurasian aspen (*Populus tremula* L.), is one of the most widely distributed tree species around the world [17]. It is a light-demanding broadleaved softwood tree species that is native to boreal zones as well as to cooler parts, such as the temperate zones of Europe and Asia [18]. It is one of the so-called pioneer species that occupy post-disturbance areas after windbreaks, fire events, or clear cutting [17,19]. Although in forest stands dominated by other species it is usually scattered, it can create homogeneous stands at post-disturbance areas [20].

The aspen grows very fast, especially during the first 15–20 years, i.e., in the period when crown competition increases [20]. Even though the commercial importance of aspen is limited, the species is often considered as a relevant part of forest ecosystems due to its fundamental importance for other plant and animal species [21,22]. This species creates conditions for existence of a variety of herbivorous, saprophytic invertebrates, fungi and lichens, birds, etc. [20]. It is one of the most attractive tree species for forage for large herbivores [23]. Hence, its presence increases the carrying capacity of red deer hunting grounds and can ensure the biological protection of other economically important tree species [24].

Moreover, the species is considered to be soil ameliorating since its foliage litter contains high concentrations of calcium, potassium, and magnesium [19]. It is also an important species for regulating the microclimate and for enhancing the structural and biological diversity of open agricultural landscapes in the temperate zone of Europe [22,25].

The share of aspen in Slovak forests is small, since from the point of its contribution to the wood stock and forest area it was ranked 18th and 16th, respectively, out of all tree species (approximately 50) recorded in the national forestry inventory [26]. According to the data from the last National Forest Inventory of Slovakia, aspen was recorded at 8% of the inventory plots [26]. Although it occurred at elevations from 200 to 1300 m a.s.l., aspen was most frequent at elevations between 450 and 550 m a.s.l. [10]. Despite its good ecological values, it is not an important tree species for any biotopes of European and national significance, nor is it a main tree species of any biotope. In Slovakia and Czechia, aspen (similarly to many other softwood broadleaved tree species) occur more frequently in stands growing on former agricultural land than on forest lands [27].

Since common aspen is fast growing and tolerant to extreme or nutrient-poor ecological conditions, it might be a prospective species to produce biomass for energetic purposes or for pulp and paper production [28]. Common aspen and its hybrids (especially *P. tremula* × *P. tremuloides*) are frequently planted in Nordic and Baltic countries [29–31] and used for pulp production. The increasing demand of woody biomass by the energy sector as well as by the forest industry will increase the need for alternative wood sources. Applying "short rotation forestry" based on fast-growing tree species, including those from the *Populus* genus, is a promising alternative [32]. While a number of papers have focused on the biomass of a variety of tree species, nearly no information on the biomass characteristics for common aspen is available and absolutely no works studied its biomass allocation patterns or component traits in terms of ecological conditions or biological aspects. Our review of the European literature focusing on common aspen indicated neglected interest in this species.

Therefore, the main aim of our paper was to quantify total tree biomass and its allocation to components in common aspen at both the tree and stand levels. A further aim of the paper was to quantify foliage traits (especially SLA) and stand canopy status (LAI) with respect to tree or stand size, specifically stem base diameter.

## 2. Material and Methods

### 2.1. Site and Stand Description

Our research focused on the forested mountainous area of central Slovakia belonging to the Western Carpathians. In general, the forest composition of the Western Carpathians, which is a part of the Carpathian Mountain Range, changes with altitude. At the lowest altitudes, oaks (mainly *Quercus robur* L. and *Q. petraeae* (Matt.) Liebl.) dominate, while at the middle altitudes European beech (*Fagus sylvatica*) is the dominant species. At higher altitudes (over approx. 900 m a.s.l.), coniferous species such as Norway spruce (*Picea abies* (L.) H. Karst.), Silver fir (*Abies alba* Mill.), Scots pine (*Pinus sylvetris* L.), and European larch (*Larix decidua* Mill.) prevail.

A preliminary selection of forest stands containing common aspen was conducted using a database of Programs of Forest Management by Stand Units in Slovakia (available on: http://gis.nlcsk.org/lgis/) based on specific information on tree species composition and stand age. The main criteria for forest selection were (1) the share of the target tree species, i.e., aspen, had to be equal to or greater than 90%, and (2) a stand age of maximum 20 years. Afterwards, we examined preselected forest stands in the field, where we checked the origin and the actual contribution of aspen to tree species composition in stands. The final selection of nine forest stands (Figure 1) was performed considering exclusively natural regeneration and nearly a 100% share of aspen. The youngest selected stand was 2 years old and the oldest stand was 15 years old. All stands were dense with no large gaps in the forest canopy.

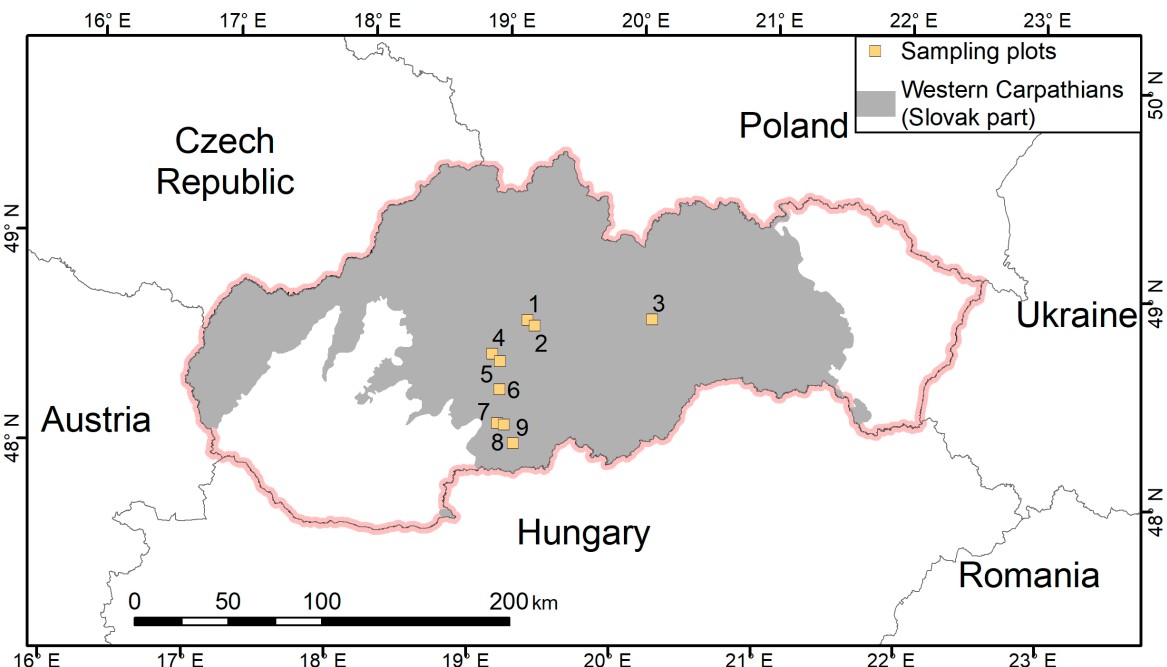

**Figure 1.** Localization of common aspen stands selected for biomass sampling (namely: 1—Podkonice I, 2—Podkonice II, 3—Telgart, 4—Kasova Lehota, 5—Straze, 6—Dobra Niva, 7—Suchan I, 8—Suchan II, 9—Opava).

The altitudinal range of the selected stands was between 335 m and 610 m a.s.l. (Table 1, i.e., they occurred in the forest vegetation zone with natural prevalence of oak and beech). In the selected region and altitudes, annual mean temperatures are between 5.0 °C and 8.5 °C, annual precipitation totals fluctuate between 600 and 900 mm, and the growing season usually lasts from 130 to 175 days. Soils are represented by Cambisols, Luvisols, and Rendzina (Table 1).

**Table 1.** Site characteristics of the forest stands selected for common aspen sampling and measurements.

| Locality Code | Name of Locality | Altitude (m a.s.l.) | Longitude (°) | Latitude (°) | Exposition (°) | Slope (%) | Soil Group |
|---|---|---|---|---|---|---|---|
| 1 | Podkonice I | 550 | 48.7932 | 19.2672 | 228 | 15 | Rendzina |
| 2 | Podkonice II | 545 | 48.7931 | 19.2671 | 230 | 16 | Rendzina |
| 3 | Telgart | 870 | 48.8359 | 20.1711 | 61 | 9 | Cambisol |
| 4 | Kasova Lehotka | 610 | 48.6204 | 19.0288 | 231 | 121 | Cambisol |
| 5 | Straze | 335 | 48.5864 | 19.0897 | 213 | 13 | Luvisols |
| 6 | Dobra Niva | 365 | 48.4522 | 19.1003 | 50 | 7 | Cambisol |
| 7 | Suchan I | 540 | 48.2897 | 19.1022 | 355 | 14 | Luvisols |
| 8 | Suchan II | 540 | 48.2896 | 19.1023 | 357 | 13 | Luvisols |
| 9 | Opava | 525 | 48.1998 | 19.2235 | 227 | 22 | Cambisol |

*2.2. Field and Laboratory Works*

Field measurements and tree samplings were performed in the second half of the growing season of 2018. In each selected stand, three circular plots were established with a distance between the plots of at least 10 m. The radius of the plots varied between 1.5 m and 3.0 m, as it depended on stand density and was chosen to cover at least 30 trees. Diameters at stem base (hereinafter as $d_0$) and tree heights were measured for all trees at the plots. In total, 27 circular plots and 971 trees were measured. Mean tree and stand characteristics for the nine forest stands are presented in Table 2. Plot-level data were used for subsequent calculations at a stand level using input data recorded at the tree level (destructive tree sampling).

Afterwards, 20 aspen trees were selected from each stand (altogether 180 individuals based on stratified random sampling) to cover all bio-sociological classes (i.e., dominant, co-dominant, sub-dominant, and suppressed). The root system of each tree was excavated to include all roots with a diameter of at least 1 mm. In addition, 5 leaves were sampled from the lower, middle, and upper thirds of the tree crowns, i.e., 15 leaves were sampled from each tree. Leaf subsamples were packed in paper envelopes identified with plot and tree codes, and the position of the leaves in the crown.

Excavated trees were cross-cut at a stem base to separate the belowground (root system) and aboveground parts. Subsequently, branches with foliage were cut off from the main stem. Stem length (tree height) and diameter $d_0$ were measured. Each tree component, i.e., root system, stems, and branches with foliage, were packed in paper bags identified with plot and tree codes. All samples were transported to the laboratory. Subsamples of leaves were scanned and their one-side leaf surface area (LA hereinafter) was calculated using the Easy Leaf Area software [33]. Each leaf was oven-dried (under 95 °C for 24 h) and weighed with a precision of 0.001 g. Individual leaf mass and area were used for the calculation of SLA at a specific crown part (upper, middle, and lower).

Leaves were manually trimmed back from the branches. Consequently, roots, stems, branches, and foliage were stored in a warm (about 28 °C), dry, and well-ventilated room. After several weeks, individual tree components were placed in the oven and dried under a temperature of 95 °C for a determined period; specifically, foliage for 48 h and woody parts (i.e., roots, stems, and branches) for 72 h to obtain dry matter. After drying, all components were weighed with a precision of 0.1 g.

**Table 2.** Stand characteristics of common aspen (mean values and standard deviations were calculated from three circular plots).

| Locality Code | Name of Locality | Mean Diameter $d_0$ (mm) | S.D. | Mean Height (m) | S.D. | Mean Canopy Closure (%) | S.D. | Mean Basal Area ($m^2$ per are) | S.D. | Mean Number of Trees (per are) | S.D. |
|---|---|---|---|---|---|---|---|---|---|---|---|
| 1 | Podkonice I | 30.4 | 1.3 | 3.29 | 0.24 | 77 | 15 | 0.347 | 0.125 | 469 | 139 |
| 2 | Podkonice II | 11.8 | 1.9 | 1.45 | 0.49 | 58 | 16 | 0.133 | 0.017 | 1241 | 240 |
| 3 | Telgart | 26.1 | 16 | 2.20 | 1.00 | 60 | 17 | 0.114 | 0.063 | 380 | 388 |
| 4 | Kasova Lehotka | 57.9 | 7.3 | 6.40 | 0.50 | 95 | 5 | 0.613 | 0.244 | 227 | 63 |
| 5 | Straze | 29.2 | 1.8 | 4.09 | 0.61 | 73 | 31 | 0.289 | 0.175 | 432 | 247 |
| 6 | Dobra Niva | 54.4 | 4.3 | 5.82 | 0.66 | 100 | 0 | 0.493 | 0.117 | 210 | 20 |
| 7 | Suchan I | 30.8 | 10.3 | 2.99 | 0.64 | 83 | 15 | 0.176 | 0.057 | 303 | 37 |
| 8 | Suchan II | 12.7 | 1.5 | 1.22 | 0.21 | 50 | 0 | 0.122 | 0.014 | 976 | 160 |
| 9 | Opava | 108.8 | 11.2 | 9.55 | 0.83 | 70 | 33 | 1.109 | 0.839 | 123 | 89 |

### 2.3. Calculations and Modelling

From the data we derived models that quantified the biomass in the foliage and woody parts of aspen trees, as well as other variables representing leaf growth (*LA, SLA, LAI, LAR, LMR*). Models were derived for several levels:

- The level of individual leaves (leaf mass $w_f$, *LA and SLA*)
- The tree level (foliage mass $w_f$, mass of woody parts $w_{wp}$, *LA, LAR, LMR*)
- The stand level (foliage mass $w_f$, mass of woody parts $w_{wp}$, *LAI*).

An allometric function with two regression coefficients ($b_1$ and $b_2$),

$$Y = b_1 d_0{}^{b_2}, \tag{1}$$

was used to describe the relationship of the particular dependent variable (represented by Y) to the $d_0$ stem base diameter. In the models for the leaf and tree levels, the tree stem base diameter $d_0$ was used, while in the stand level models a mean stand diameter at the stem base $d_{0g}$ was used. The frequently used diameter at breast height was not applicable in our analyses, because some of the selected trees did not reach a height of 1.3 m. Almost all constructed equations were derived in a basic power form. The equations of foliage and wood biomass were derived as linearised logarithmic allometric equations that were subsequently transformed back to the exponential form. This approach was applied to ensure a methodological link to previous works that quantified the biomass of components in young trees [34]. The shape of the allometric function after the logarithmic transformation and its back transformation is

$$Y = e^{(b_1 + b_2 \cdot \ln d_0)} \cdot \lambda \tag{2}$$

where $\lambda$ is a correction factor.

A logarithmic version of the equation is frequently used to exclude heteroscedasticity of residues, which is always present in the calculation of biomass components. Using a logarithmic equation enables the estimation of parameters with linear regressions, which fulfils the assumption of constant variance of residues. Although in the last years non-linear regression methods have been developing, the opinions on these two methodological approaches differ between papers, e.g., [35–37].

Models at a level of individual leaves were derived for leaves at different crown parts (upper, middle, and lower) using measured data of $d_0$ (mm), $w_f$ (g), and *LA*. *SLA* was calculated using the following formula

$$SLA = \frac{LA}{w_f} \tag{3}$$

Since only $d_0$ diameter and $w_f$ and $w_{wp}$ mass were measured for a tree, leaf area *LA* of the whole tree had to be calculated otherwise. In our work we derived a new model from the measured foliage mass of a particular tree and a tree *LA* calculated as an arithmetical average from the LA values of nine leaves taken from three crown parts of the particular tree using the equation

$$LA = w_f \frac{\sum_{i=1}^{n} \left( \frac{LA_i}{w_{f_i}} \right)}{n} \tag{4}$$

where:

*LA*—leaf area of a tree (cm$^2$);
$LA_i$—leaf area of *i*th sampled leaf (cm$^2$);
$w_f$—mass of all tree leaves (g);
$w_{f_i}$—mass of *i*th sampled leaf (g);
*n*—number of leaves sampled from one tree (i.e., 15).

For the models at a stand level, mean stand diameter $d_{0g}$ (mm) was calculated as a quadratic mean of all tree diameters using the formula

$$d_{0g} = \sqrt{\frac{\sum\limits_{i=1}^{n} d_i^2}{n}} \tag{5}$$

Leaf area index *LAI* was calculated as the *LAI* under the assumed full canopy coverage ($LAI_{100\%}$) using the relationship

$$LAI_{100\%} = \frac{100 \sum\limits_{i=1}^{n} LA_i}{Sc} \tag{6}$$

where:

$LA_i$—leaf area of $i$th tree at a plot (m$^2$ m$^{-2}$);
$S$—plot area in m$^2$;
$C$—crown canopy coverage at a plot in %;
$n$—number of trees at a plot.

Similarly, variables in other models (volume of woody parts, and foliage volume) were also converted to 100% crown canopy.

Finally, a generalised linear mixed model (GLMM) was created to evaluate the combined influence of $d_0$ and position of the leaves in the crown on leaf traits. The model has the following form:

$$y = \alpha + \beta \times \ln(d_0) + \gamma_1 \times M + \gamma_2 \times U \tag{7}$$

where:

$y$—a dependent variable, namely leaf mass, leaf area, or specific leaf area;
$d_0$—diameter at stem base;
$M$—a dummy variable that represents the middle crown part, i.e., if the leaf was taken from the middle part, the value is 1, otherwise it is 0;
$U$—a dummy variable that represents the upper crown part;
$\alpha$, $\beta$, $\gamma_1$ and $\gamma_2$—the regression coefficients of the model.

All statistical analyses were performed in Statistica 10.0 and R software version 3.5.1 [38].

## 3. Results

Heights and stem base diameters of the sampled trees varied between 0.4 m and 10.5 m and from 3.3 mm to 100.9 mm, respectively (Figure 2; Table 3). The relationship between $d_0$ diameter and tree height was best described by a fractional power relationship, specifically, $h = \frac{d_0^2}{8.221 + 7.077 d_0 + 0.021 d_0^2}$, which explained almost 90% of the variability. This relationship may help users who prefer using tree height as an independent variable in biomass models, to convert our models that are based on $d_0$ diameter.

The values of the measured leaves revealed high variation in all the assessed leaf characteristics, i.e., leaf mass (from 0.019 to 0.513 g), leaf area (between 3.79 and 50.81 cm$^2$) as well as SLA (71.33 to 374.24 cm$^2$ g$^{-1}$; Table 4). Leaf mass of individual leaves differed along the vertical crown profile, with the heaviest leaves in the upper crown part and the lightest ones in the bottom part of the crown (Figure 3a). At the same time, leaf area increased with tree size as represented by $d_0$ diameter (Figure 3b), although differences between the crown parts were less evident than in leaf mass. The influence of leaf position in the crown and tree size on SLA was very evident (Figure 3c). The largest SLA values were found for small trees and lower crown parts. Moreover, differences in SLA between the crown parts increased with tree size.

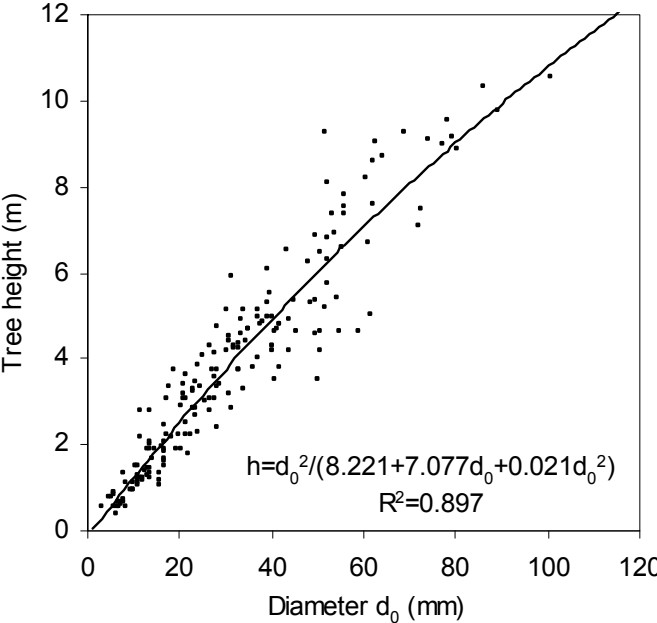

**Figure 2.** Relationship between $d_0$ stem base diameter and tree height derived from the sampled common aspen trees (values of SE were 11.812, 0.693, and 0.009, and the *p*-values 0.487, < 0.001, and 0.017 for parameters $b_1$, $b_2$, and $b_3$, respectively; MSE = 0.612).

**Table 3.** Descriptive statistics of the sampled common aspen trees.

| Tree Parameter (unit) | Mean | S.D. | Min. | Max. | 25th Percentile | 75th Percentile |
|---|---|---|---|---|---|---|
| Diameter $d_0$ (mm) | 31.4 | 20.0 | 3.3 | 100.9 | 15.5 | 43.1 |
| Height (m) | 3.81 | 2.42 | 0.40 | 10.54 | 1.87 | 4.96 |
| Roots biomass (g) | 262 | 376 | 1 | 2190 | 23 | 310 |
| Stem biomass (g) | 700 | 1293 | 1 | 8352 | 30 | 674 |
| Branches biomass (g) | 251 | 460 | 0 | 2770 | 32,813 | 284 |
| Foliage biomass (g) | 94 | 131 | 1 | 737 | 33,878 | 129 |
| Woody parts biomass (g) | 1397 | 2448 | 12,145 | 14,995 | 69 | 1391 |
| Whole tree biomass (g) | 1456 | 2530 | 45,809 | 15,651 | 83 | 1538 |

**Table 4.** Statistical characteristics of leaf traits of the sampled common aspen trees.

| Leaf Trait (unit) | Crown Part | Mean | S.D. | Min. | Max. | 25th Percentile | 75th Percentile |
|---|---|---|---|---|---|---|---|
| Leaf mass (g) | Upper | 0.171 | 0.079 | 0.029 | 0.513 | 0.114 | 0.222 |
| | Middle | 0.129 | 0.063 | 0.029 | 0.347 | 0.080 | 0.168 |
| | Lower | 0.101 | 0.058 | 0.019 | 0.329 | 0.054 | 0.130 |
| Leaf area (cm$^2$) | Upper | 21.50 | 7.29 | 6.33 | 44.63 | 16.32 | 25.50 |
| | Middle | 19.71 | 7.73 | 8.20 | 50.27 | 14.11 | 23.84 |
| | Lower | 17.42 | 8.10 | 3.79 | 50.81 | 11.98 | 21.76 |
| SLA (cm$^2$ g$^{-1}$) | Upper | 138.88 | 42.68 | 71.33 | 308.17 | 108.16 | 159.40 |
| | Middle | 167.31 | 47.01 | 91.50 | 374.24 | 131.90 | 193.70 |
| | Lower | 190.78 | 49.85 | 97.25 | 355.21 | 152.06 | 227.78 |

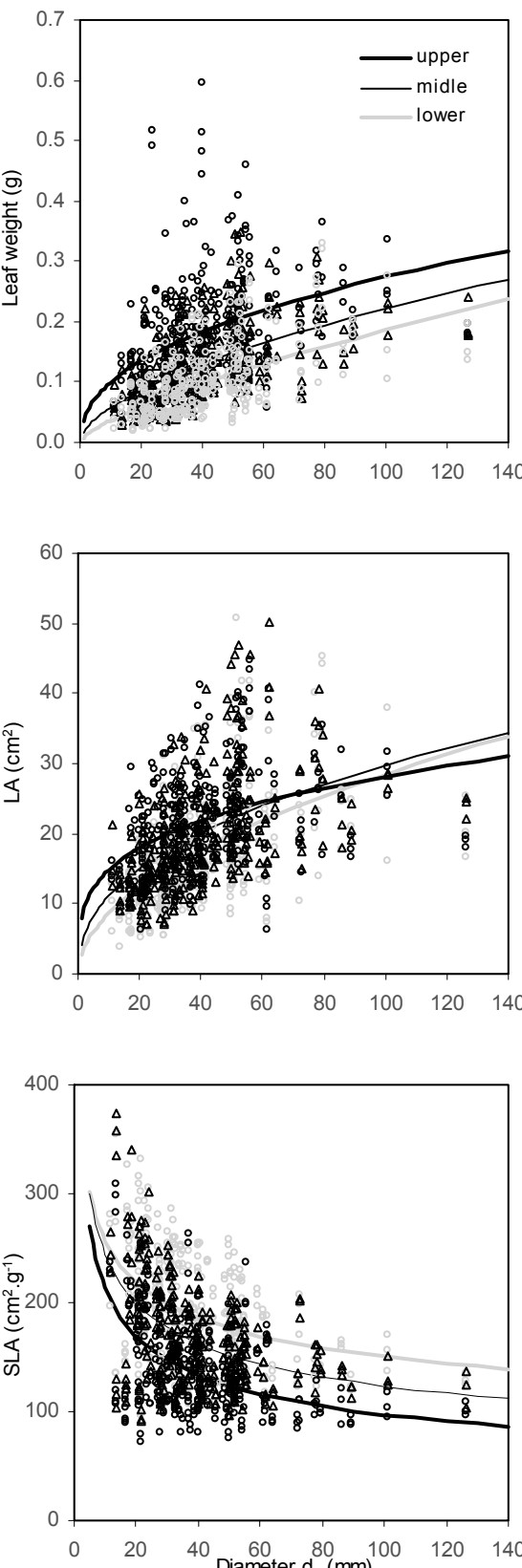

**Figure 3.** Relationship between $d_0$ stem base diameter and individual leaf mass (upper plate), leaf area (middle plate), or specific leaf area (lower plate) in the sampled common aspen trees with respect to foliage position along the tree crown.

Our mixed models (Table 5) also revealed a significant influence of the crown part from which the leaves were sampled. The models confirmed that both leaf mass and leaf area increase from the bottom to the top of the crown, i.e., leaves in the upper crown part have larger values of area and mass than in the middle part, which has larger and heavier leaves than the lower part. In the case of specific leaf area, we found an opposite order, i.e., lower > middle > upper, which means that the specific leaf area of the bottom leaves is greater than the SLA of the middle and upper ones (leaves on the top of the crown can be considered as those with minimum SLA, given the diameter at the base is equal for the samples).

**Table 5.** Results of the generalised linear mixed models derived for leaf traits based on Equation (7).

| Leaf Trait (unit) | GLMM Component | Estimate | Standard Error | t-Value | *p*-Value | Significance |
|---|---|---|---|---|---|---|
| | Intercept | −0.195 | 0.0167 | −11.695 | 0.000 | *** |
| | $Ln(d_0)$ | 0.082 | 0.0045 | 18.119 | 0.000 | *** |
| Leaf mass (g) | Middle crown part | 0.082 | 0.0050 | 5.665 | 0.000 | *** |
| | Upper crown part | 0.079 | 0.0050 | 15.797 | 0.000 | *** |
| | *$R^2$ = 0.625, AIC = −2606* | | | | | |
| | Intercept | −12.863 | 1.764 | −7.292 | 0.000 | *** |
| | $Ln(d_0)$ | 8.391 | 0.478 | 17.560 | 0.000 | *** |
| Leaf area ($cm^2$) | Middle crown part | 2.301 | 0.526 | 4.374 | 0.000 | *** |
| | Upper crown part | 4.020 | 0.529 | 4.374 | 0.000 | *** |
| | *$R^2$ = 0.725, AIC = 6468* | | | | | |
| | Intercept | 359.55 | 10.866 | 33.088 | 0.000 | *** |
| | $Ln(d_0)$ | −46.77 | 2.944 | −15.889 | 0.000 | *** |
| SLA ($cm^2\ g^{-1}$) | Middle crown part | −23.48 | 3.247 | −7.233 | 0.000 | *** |
| | Upper crown part | −52.51 | 3.244 | −16.183 | 0.000 | *** |
| | *$R^2$ = 0.654, AIC = 10089* | | | | | |

Note: *** significant at 99.9% significance level.

Allometric biomass models for individual tree components as well as for whole-tree biomass showed their close dependences on $d_0$ diameter (in all cases $p < 0.001$; see Table 6). Fitted values showed that while aspen trees with a diameter of 50 mm had about 2.5 kg of woody biomass and 0.2 kg of foliage biomass, individuals with a diameter of 100 mm had 16.4 kg of biomass in woody parts and 0.7 kg in foliage (Figure 4). This means that biomass in woody parts increased with tree size faster (in the $d_0$ interval 50–100 mm as much as 6.6 times) than in foliage (in the diameter interval 50–100 mm only by 3.5 times). If foliage area at a tree level is considered, increase in the $d_0$ interval from 50 mm to 100 mm was from 2.6 $m^2$ to 8.7 $m^2$ (tripled value; Figure 5). As for LAR and LMR, both indicators decreased with tree size (Figure 6; Table 7). This fast decrease was typical in very small trees (with $d_0$ below approximately 20 mm); then the trend became milder.

**Table 6.** Biomass models of individual tree components in common aspen using stem base diameter as an independent variable, showing their regression coefficients ($b_1$, $b_2$), standard errors (SE), *p*-values (*P*), coefficients of determination ($R^2$), mean squared errors (MSE), logarithmic transformation bias ($\lambda$), and standard deviation (SD) (see Equation (2)).

| Biomass of Tree | $b_1$ | S.E. | *P* | $b_2$ | S.E. | *P* | $R^2$ | MSE | $\lambda$ | S.D. |
|---|---|---|---|---|---|---|---|---|---|---|
| Components (g) | | | | | | | | | | |
| Roots (A) | −3.315 | 0.130 | <0.001 | 2.410 | 0.039 | <0.001 | 0.954 | 0.145 | 1.078 | 0.474 |
| Stem (B) | −3.612 | 0.107 | <0.001 | 2.795 | 0.033 | <0.001 | 0.976 | 0.098 | 1.048 | 0.323 |
| Branches (C) | −5.683 | 0.196 | <0.001 | 3.010 | 0.060 | <0.001 | 0.935 | 0.306 | 1.116 | 0.710 |
| Leaves (D) | −2.907 | 0.225 | <0.001 | 2.020 | 0.069 | <0.001 | 0.829 | 0.429 | 1.191 | 0.634 |
| Woody parts (A+B+C) | −2.760 | 0.093 | <0.001 | 2.699 | 0.028 | <0.001 | 0.982 | 0.071 | 1.036 | 0.293 |
| Whole tree (A+B+C+D) | −2.379 | 0.092 | <0.001 | 2.618 | 0.028 | <0.001 | 0.981 | 0.070 | 1.036 | 0.293 |

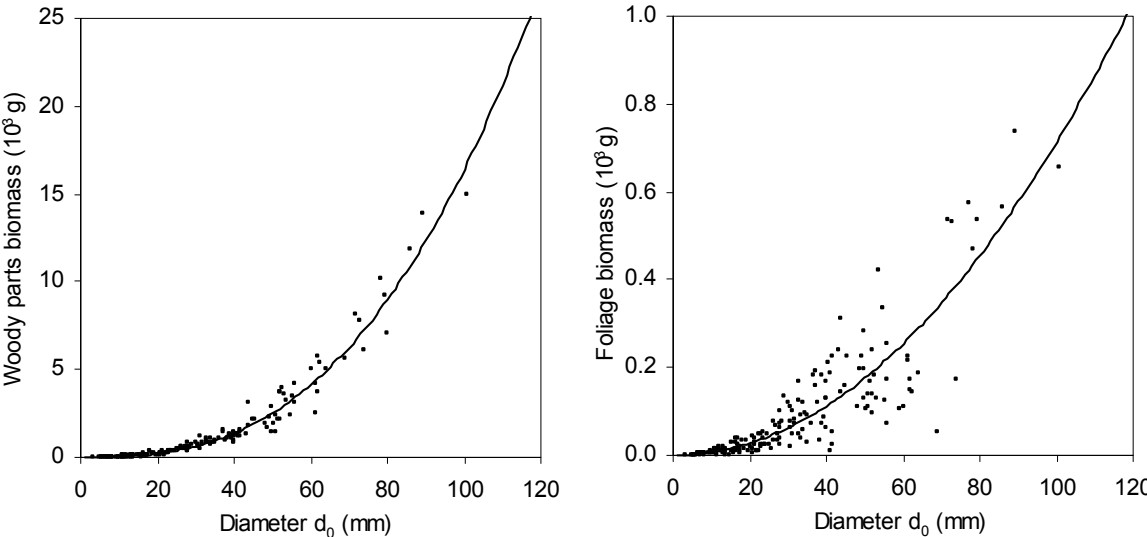

**Figure 4.** Biomass of woody parts (left plate) and foliage (right plate) in the sampled common aspen trees plotted against the $d_0$ stem base diameter (see also Table 6).

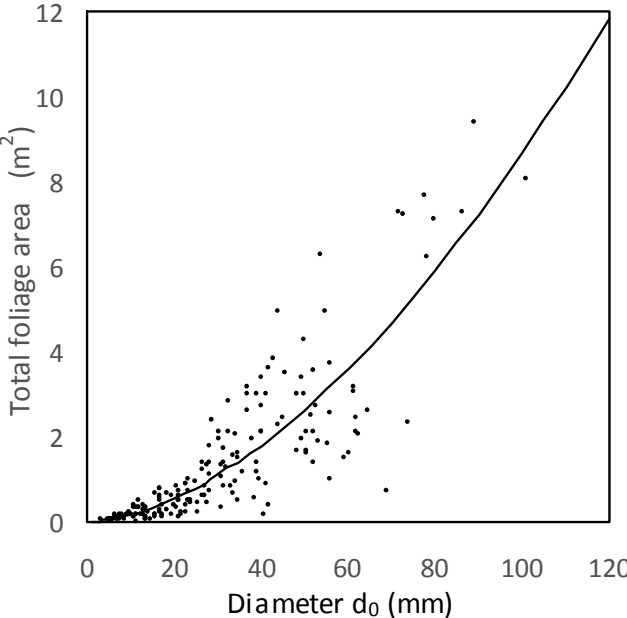

**Figure 5.** Total tree foliage area of the sampled common aspen trees against the $d_0$ stem base diameter (see also Table 7).

**Table 7.** Models for calculating tree foliage area, leaf area ratio (LAR), and leaf mass ratio (LMR) in common aspen using stem base diameter as an independent variable, showing their regression coefficients ($b_1$, $b_2$), standard errors (S.E.), $p$-values ($P$), coefficients of determination ($R^2$), and mean squared errors (MSE) (see Equation (1)).

| Tree Characteristics (unit) | $b_1$ | S.E. | $P$ | $b_2$ | S.E. | $P$ | $R^2$ | MSE |
|---|---|---|---|---|---|---|---|---|
| Tree foliage area ($cm^2$) | 0.0031 | 0.0012 | 0.010 | 1.723 | 0.093 | <0.001 | 0.734 | 0.834 |
| LAR ($cm^2\ kg^{-1}$) | 308.4 | 37.2 | <0.001 | −0.869 | 0.051 | <0.001 | 0.634 | 165.93 |
| LMR ($kg\ kg^{-1}$) | 0.657 | 0.083 | <0.001 | −0.585 | 0.048 | <0.001 | 0.458 | 1.297 |

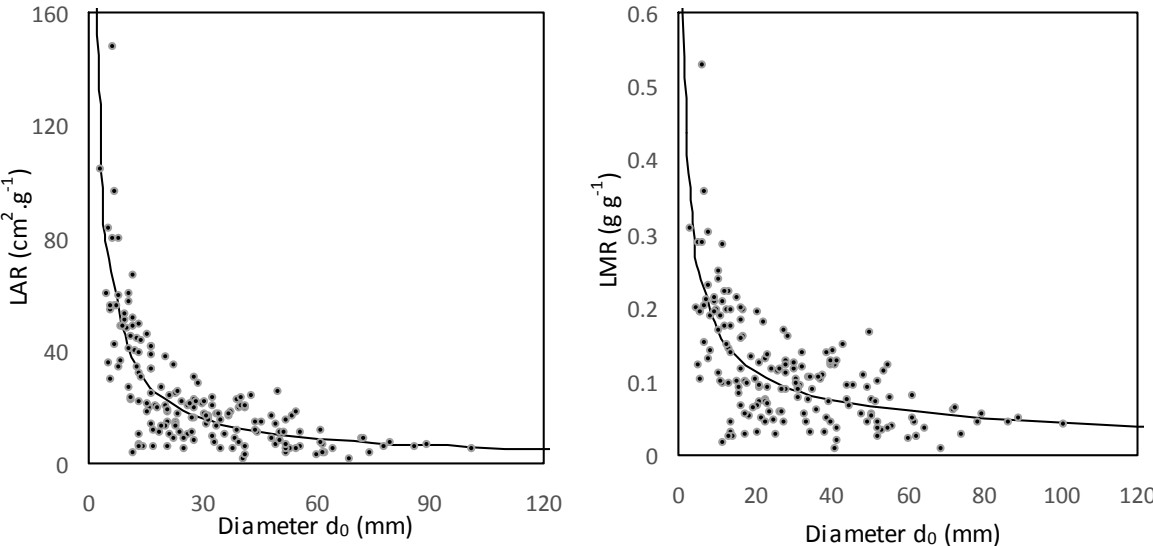

**Figure 6.** Leaf area ratio (left plate) and leaf mass ratio (right plate) plotted against the $d_0$ stem base diameter of the sampled common aspen trees (see also Table 3).

Finally, quantities of woody parts and foliage were estimated at the stand level. The fitted curves (Figure 7 and Table 8) showed that the increase in stand biomass in woody parts with an increasing $d_0$ was greater than the increase in foliage biomass. For instance, while an aspen stand with a mean diameter $d_0$ of 50 mm had about 8 kg of biomass in woody parts per $m^2$, a stand with a $d_0$ of 100 mm contained about 32 kg of woody biomass per $m^2$. This represents a fourfold increase. On the other hand, the difference between the aspen stands with mean diameters $d_0$ of 50 mm and 100 mm was only doubled (0.5 kg $m^{-2}$ versus 1.2 kg $m^{-2}$). Values of LAI increased with $d_0$ first fast, and then more slowly (Figure 8, Table 8). For instance, while a stand with $d_0$ equal to 50 mm had an LAI of 7.3 $m^2 m^{-2}$, the LAI of a stand with a $d_0$ of 100 mm was 10.9 $m^2 m^{-2}$. The greatest value of LAI equal to 13.5 was found in the aspen forest with a mean stand diameter $d_0$ of 120 mm.

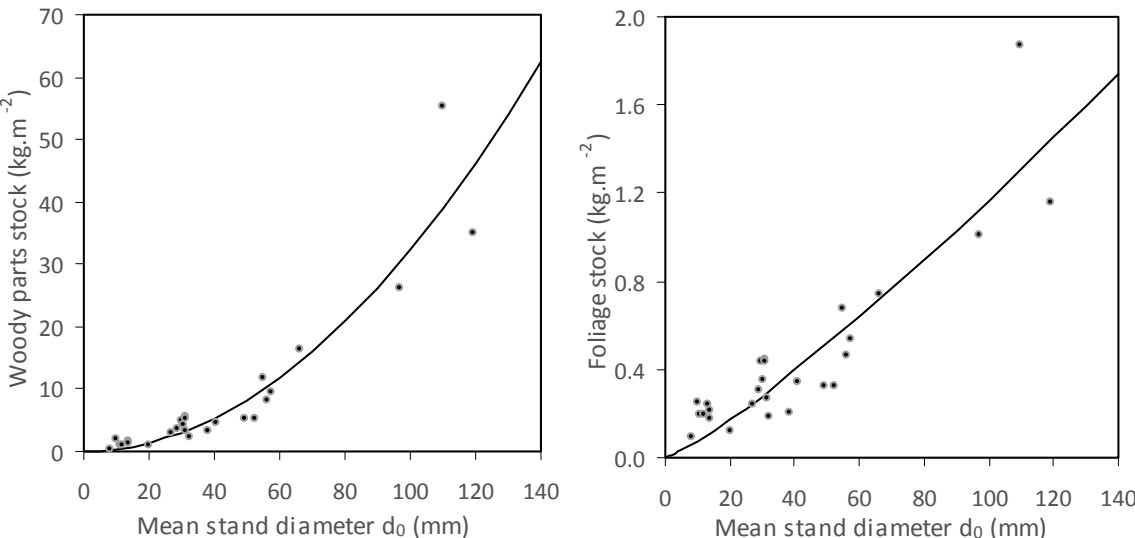

**Figure 7.** Stand biomass of woody parts (left plate) and foliage (right plate) in common aspen fully-stocked stands plotted against the $d_0$ mean stand diameter at the stem base (see also Table 8).

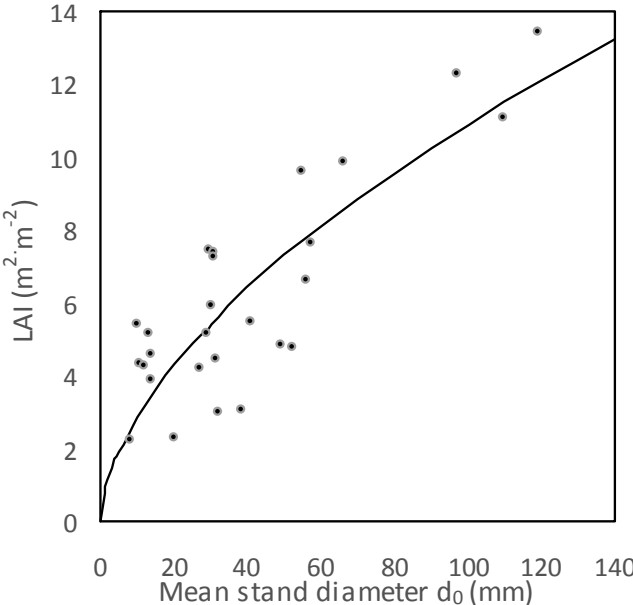

**Figure 8.** Leaf area index of fully-stocked common aspen stands plotted against the $d_0$ mean stand diameter at the stem base (see also Table 8).

**Table 8.** Models for calculating the biomass of woody parts, foliage, and leaf area index in common aspen stands using stem base diameter as an independent variable, showing their regression coefficients ($b_1$, $b_2$), standard errors (S.E.), *p*-values (*P*), coefficients of determination ($R^2$), and mean squared errors (MSE) (see Equation (1)).

| Stand Characteristics (unit) | $b_1$ | S.E. | *P* | $b_2$ | S.E. | *P* | $R^2$ | MSE |
|---|---|---|---|---|---|---|---|---|
| Woody parts stock (kg m$^{-2}$) | 0.0039 | 0.0036 | 0.291 | 1.959 | 0.201 | <0.001 | 0.882 | 18.497 |
| Foliage stock (kg m$^{-2}$) | 0.005 | 0.003 | 0.082 | 1.184 | 0.125 | <0.001 | 0.810 | 0.029 |
| Leaf area index (m$^2$ m$^{-2}$) | 0.929 | 0.305 | 0.005 | 0.522 | 0.082 | <0.001 | 0.616 | 3.160 |

## 4. Discussion

### 4.1. Foliage- and Tree-Level Traits

Foliage traits are important characteristics that specify the rate of absorbing photosynthetically active radiation used in photosynthesis (leaf surface area), the metabolic costs of leaf construction (leaf mass), and the efficiency with which the leaf captures light relative to the biomass invested in the plant biomass (leaf mass ratio) or leaf (specific leaf area). Their values obtained from field measurements are required not only for the assessment of the trade-off between ecosystem mass and energy balance, but also for accurate parametrisation and predictions of forest growth by process-based models [39]. Increasing mass and area of single leaves with tree size was typical for young common aspen trees, while the opposite situation was found for SLA. Besides tree size, the position of the leaves along the vertical crown profile appeared to be important for foliage properties, more clearly for leaf mass and SLA than leaf area.

Our study indicated that while smaller trees had larger (as for area) leaves in the upper crown part and smallest in the lower crown part, the situation in bigger trees was reversed. Larger leaves in the upper crown part were also found for European beech trees with a $d_0$ diameter below 60 mm [40]. Similarly, [41] showed that shade leaves in young beech stands had prevailingly lower leaf areas than sunlit leaves. A reverse situation was recorded in old beech stands [42,43]. The increase of SLA from the top to the bottom of the crown was demonstrated also in other works [40,42,44,45]. It seems that SLA is a better indicator of foliage plasticity to environmental factors, especially light conditions [42,46], than the area of an individual leaf.

Our results showed that the SLA of common aspen leaves fluctuated between approx. 100 and 300 cm$^2$ g$^{-1}$. Other published values of SLA for common aspen varied from 39 to 182 cm$^2$ g$^{-1}$ [4], i.e., they were lower than most of our values (Figure 3c) although they represented stands of similar ages (from 5 to 24 years). The results from European beech stands showed SLA values similar to our results, ranging from 80 to 480 cm$^2$ g$^{-1}$ [41,42,44,47,48].

The values of LMR for small aspen trees were high and sharply decreased with tree size. The LMR of aspen trees with a $d_0$ diameter of about 20 mm was below 0.1, while the LMR of trees with a $d_0$ of 80 mm reached a value of only about 0.05. Our previous study [49] for *Fagus sylvatica* and *Quercus petraea* showed that the LMR of trees with a $d_0$ between 20 and 80 mm decreased from 0.2 to 0.1 and from 0.15 to 0.10, respectively. These values indicate that in the investigated diameter interval, the contribution of foliage to total biomass of *Fagus sylvatica* and *Quercus petraea* was double the foliage contribution in aspen. This suggests a high growth efficiency of common aspen in tree biomass production [50].

Our height–diameter model for young common aspen trees showed a steep, almost a linear increase in tree height with increasing $d_0$ (Figure 2). Since we have previously constructed height–diameter models for other tree species [51], we were able to analyse inter-specific differences. We compared aspen versus seven other broadleaved tree species, i.e., *Fagus sylvatica*, *Carpinus betulus*, *Quercus petraea*, *Acer pseudoplatanus*, *Fraxinus excelsior*, *Salix caprea*, and *Sorbus aucuparia*. The comparison showed that aspen trees were higher than the trees of other species with the same $d_0$ diameter. For instance, while aspen with a $d_0$ equal to 80 mm was approximately 9 m tall, the height of *Acer pseudoplatanus* was only 8 m, *Carpinus betulus* was 7 m, and all other species were less than 7 m tall. This indicates that in comparison with other species young aspen trees invest more carbohydrates into height increment than into radial stem increment. From the point of inter-specific competition for light, that might be its advantage in young mixed stands. In this context, for instance, [31] pointed out the very high competitive ability of aspen, especially on fertile soils with fresh moisture conditions.

The calculated mean woody biomass value of the investigated common aspen trees was lower than the published values from young hybrid aspen stands (see e.g., [31,52,53]) or other poplar cultivars [54] of similar ages, while the range of our values was usually greater than in other studies. This was expected since our trees originated from natural regeneration, while the other studies dealt with trees from plantations established for biomass production.

*4.2. Stand Level Traits*

Mean woody stand biomass stock of young common aspen at our plots was 8.20 ± 12.30 kg/m$^2$. The value does not significantly differ from other published values, e.g., [31] reported an average value of aboveground woody biomass for his hybrid aspen stands equal to 13.5 ± 5.3 kg/m$^2$. Similar to tree biomass, the variability in stand biomass between plots was large (Figure 7). A wide range of aboveground stand biomass values was also reported in other studies, e.g., the aboveground biomass of hybrid aspen stands in Sweden ranged from 1.43 to 21.9 kg/m$^2$ [31,55], or young common aspen stands aged below 20 years included in the Eurasian database [56,57] were characterised by aboveground biomass values ranging from 1.6 to 10.99 kg/m$^2$. Faster biomass accumulation has been documented at fertile sites with good soil conditions [30].

Stand biomass stock of woody tree parts in young common aspen forests exponentially increased with increasing stand dimensions (mean stand diameter $d_0$), while foliage biomass stock increased almost linearly (Figure 7). Hence, the ratio between woody part biomass and foliage biomass in an aspen forest with a mean stand stem base diameter of 50 mm was approximately 20, the same ratio was almost 30 if the mean stand diameter was 100 mm. These changes with stand development reflect not only the physiological (increasing growth efficiency) but also production–ecological aspects of forests, especially their carbon sequestration potential. Woody parts represent long-term carbon cycling components, while leaves store carbon only for a short time. Approximately 1/20 of total tree biomass annually falls off on the ground in young aspen forests with a mean stand diameter of 50 mm, while in the stand with a diameter of 100 mm it is only 1/30 (here we obviously neglected tree mortality as a source of carbon

transfer from biomass to necromass). Foliage litter was estimated to be about 0.5 kg per m$^2$ and 1.1 kg per m$^2$ in stands with a $d_0$ mean diameters of 50 mm and 100 mm, respectively. A similar decrease in foliage contribution to total tree biomass has been recorded in beech [58] or birch stands [1].

The LAI of our aspen stands fluctuated between 2 m$^2$ m$^{-2}$ and 14 m$^2$ m$^{-2}$. The maximum estimated value of nearly 14 m$^2$ m$^{-2}$ is greater than the values published in most of the other papers covering a variety of forest tree species (often European beech) [46,47,59–63]. The prevailing part of the other papers presented LAI values up to 10 m$^2$ m$^{-2}$. In fact, the results of most papers originated from older growth stages than in our case. The few papers that showed LAI values near 14 m$^2$ m$^{-2}$ [64–67] originated from young growth stages. This might suggest that forest stands reach their maximum LAI in the young stages and after that the values decrease with stand development, which has already been documented for Norway spruce [65]. Our results bring new findings about biomass allocation and foliage traits of common aspen at both the tree and stand levels. Since this kind of study has been missing for common aspen, our results might not be compared or synthesized with other knowledge. To obtain biomass information about less abundant and/or currently commercially unimportant species, foresters often use equations adopted from the already published sources, or those developed for other species. However, such an approach may cause a significant bias in the obtained estimates, since growing conditions and tree properties vary across the regions and country-specific models also differ from each other [67]. Comparison of different models for birch showed that non-local models may overestimate total aboveground biomass of thin trees with a diameter at breast height below 4 cm [68]. Moreover, equations derived 20 years before or more may no longer be valid due to the recent changes in environmental conditions [69]. Hence, model updates are required even for thoroughly studied commercial species.

## 5. Conclusions

This study brings novel findings for a very productive tree species with modest ecological demands, but competitive to other plant species (both weeds and trees). Therefore, it might be planted on sites which are not attractive and/or suitable for traditionally important commercial tree species (e.g., European beech and Norway spruce). Moreover, the species could be very useful especially for energy production. A rapid increase in bioenergy demands has been predicted for most future scenarios, considering especially ecological (e.g., global carbon balance) and technological issues [70]. Biomass of short-living tree species may significantly contribute to bioenergy production and carbon sequestration [71]. Therefore, we believe that our results might be significant for both theoretical (e.g., biomass production and partitioning modelling) and practical (forestry and agro-forestry stakeholders) purposes.

**Author Contributions:** Conceptualization, B.K.; data curation, J.P. and V.Š.; funding acquisition, B.K.; investigation, B.K., J.P. and V.Š.; methodology, B.K. and P.S.; visualisation, J.P. and V.Š; supervision, B.K.; writing—original draft, B.K., J.P., V.Š., P.S. and K.M.; writing—review and editing, B.K. and K.M. All authors have read and agreed to the published version of the manuscript.

**Funding:** This research was funded by grant "EVA4.0", No. CZ.02.1.01/0.0/0.0/16_019/0000803 financed by OP RDE, also by the projects APVV-0584-12 and APVV-18-0086 from the Slovak Research and Development Agency as well as by the project "Research and innovation for supporting competitiveness of the Slovak forestry sector" (SLOV-LES) financed by the Ministry of Land Management and Rural Development of the Slovak Republic.

**Conflicts of Interest:** The authors declare no conflict of interest.

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
