# Peer review of "Biomass Allocation into Woody Parts and Foliage in Young Common Aspen (Populus tremula L.)—Trees and a Stand-Level Study in the Western Carpathians"

_forests, doi:10.3390/f11040464_

Round 1

Reviewer 1 Report

  1. Research design is generally appropriate but can be improved. The criteria for selecting research material are clear but the number of sample stands is quite low (9), moreover authors did not included any statistical description of stands and sample trees, so the variablity of the biometric features within the stands and plots cannot be judged and possibly can have some impact on the results.
  2. The methods used in study are well described and clear but the constructed models are quite basic. Unfortunately authors don't explain why they are not trying to incorporate more appropriate methods. One of the most important improvement would be introduction of coding the upper, middle and lower part of the crown as dummy variables. It will simplyfy the results into one leaf model instead of three and give the opportunity to judge the differences between those three parts on the basis of statistical testing instead of chart assessment. Even more important improvement for all constructed models could be introduction of the mixed models approach, which would give the opportunity to include autocorrelation between trees from the same stand. The used approach in the study treats all trees as independent which is undoubtly false - trees from the same site tends to have similar allometric relations and such autocorrelation should be included with the help of hierarchical modelling. Also the leaves from the different parts of the crown probably are autocorrelated.
  3. The models for the biomass of the woody parts and biomass of foliage are well developed and decribed but the discussed changes along with the changing diameter can be much clearer if the simple ratio between woody parts and foliage will be calculated, modelled and discussed. 
  4. Designed models for biomass, woody part stock and foliage stock have obvious signs of heteroscedasticity. Authors do not mention the used method of parametres computation so it can be assumed that it was Ordinaly Least Squares (OLS) approach while the weighted approach would be more appropriate.

The results will be probably much clearer if the mentioned above changes will be introduced - the models will be coherent and their interpretation will be easier. Although all above-mentioned corrections would be very welcomed, the paper even in the actual form present original and very interesting results. Statistical analysis can be improved but the presented models are properly constructed and analysed. In my opinion the biomass allocation of common aspen, even with actual low importance of this species, should be recognized and described. Presented study is fullfilling this need in proper way and I assume that presented study can be very interesting for the readers of "Forests" journal.

Author Response

  1. Research design is generally appropriate but can be improved. The criteria for selecting research material are clear but the number of sample stands is quite low (9), moreover authors did not included any statistical description of stands and sample trees, so the variablity of the biometric features within the stands and plots cannot be judged and possibly can have some impact on the results.

Thank you for the comment. It is true that the number of sampled stands is not high, however, in each stand 3 plots were established, i.e. in total we had 27 plots and we measured nearly 1000 trees, which is a reasonable number. The extent of sampling and measurements is always a result of a compromise between time (expanses) consumption and accuracy.

We added four tables with the description of site conditions, sampled trees and foliage, and the information about the measured plots.

  1. The methods used in study are well described and clear but the constructed models are quite basic. Unfortunately authors don't explain why they are not trying to incorporate more appropriate methods. One of the most important improvement would be introduction of coding the upper, middle and lower part of the crown as dummy variables. It will simplyfy the results into one leaf model instead of three and give the opportunity to judge the differences between those three parts on the basis of statistical testing instead of chart assessment. Even more important improvement for all constructed models could be introduction of the mixed models approach, which would give the opportunity to include autocorrelation between trees from the same stand. The used approach in the study treats all trees as independent which is undoubtly false - trees from the same site tends to have similar allometric relations and such autocorrelation should be included with the help of hierarchical modelling. Also the leaves from the different parts of the crown probably are autocorrelated.

Thank you for the comment. We derived a generalized linear mixed model with dummy variables for different crown parts as suggested by the reviewer and added the results into the manuscript in Table 6.

The models for the biomass of the woody parts and biomass of foliage are well developed and decribed but the discussed changes along with the changing diameter can be much clearer if the simple ratio between woody parts and foliage will be calculated, modelled and discussed. 

We used LMR – proportion between foliage mass to total tree mass. This indicator is generally respected and recognized in biology, therefore we preferred LMR to ratio between foliage biomass and woody parts biomass.      

  1. Designed models for biomass, woody part stock and foliage stock have obvious signs of heteroscedasticity. Authors do not mention the used method of parametres computation so it can be assumed that it was Ordinaly Least Squares (OLS) approach while the weighted approach would be more appropriate.

We agree that heteroscedasticity is a general problem for modeling biological phenomena. Weighted approach might partly solve this issue, however we used OLS – since it has been the most common approach used in the literature (including many papers which were previously published in top forestry journals).  

Reviewer 2 Report

1.The authors must a comparative study at the level of biomass in Europe.

2.At least 10 references bibliography in 2016-2019 year needed.

Author Response

The quality of English language

The text was carefully reviewed and edited once again rewritten by Dr. Katarina Merganicova with Cambridge Proficiency Certificate to improve English.

  1. The authors must a comparative study at the level of biomass in Europe.

Thank you for the comment. We added a comparison with published values on tree and stand biomass as well as other characteristics in Discussion.  

  1. At least 10 references bibliography in 2016-2019 year needed.

To fulfill the query of the reviewer we added extra text in the Introduction section as well as in the Discussion section. In total, we added 17 related references, most of them published in the last three years.

Reviewer 3 Report

The overall quality of the paper is average and it has some statistical issues that should be solved.

The introduction and the presentation of experimental design are quite complete. Nevertheless, some questions arise: the use of diameter at stem base (do) is not properly justified in the paper. Why dou you use it instead of DBH? Saying we are dealing with young stands should not be enough. Is there any scientific reason to use do over DBH?

Some of the results are weak and others do not have the correct explanation:

The equation relating do and tree height has no statistical diagnosis except the R2. Authors should show this diagnosis (bias, RMSE, basic statistical problems...) if they are suggesting other authors to apply in order to use tree height as independent variable in biomass alometric equations. Error additivity should be taken into account.

Basic field data tables are missing. Even if some of the ranges are showed in the text, a table with max, min, mean and number of observations for each variable would help understand the results.

Table 1: R2 are very poor. Even if parameters are significant, a broader statistical analysis is needed. Models with R2 of 0.09 to 0.34 do not explain at all the variability that may occur. Even accepting some of the less poor R2s, more robust statistical analysis is needed (residues homocedasticity, normality, VIF, RMSE...etc.)

Table 3-Table 4: Some parameters are not significant (or slightly significant) with big MSEs in some cases.

In my opinion, overally bigger effort is needed in modelling procedures in order to avoid the shown problems of the models. Besides, there is no explanation of low R2s or non-significance of some parameters. Nevertheless, more modelling effort is needed.

The discussion correctly shows the analysis of the results in terms of allocation, even if those could be shown as trends also in results.

Some of the assumptions must be taken with care: The point of maturity the trees reach the maximum LAI should not be assumed as comparison with other species and locations. The lack of data for adult trees in the species is not enough reason to assume some comparisons.

In general, I find the study interesting, with a good introduction, methodology design and a correct discussion. Nevertheless, all the modelling procedures should be improved, together with the wway of exposing the results.

Author Response

The overall quality of the paper is average and it has some statistical issues that should be solved.

The introduction and the presentation of experimental design are quite complete. Nevertheless, some questions arise: the use of diameter at stem base (do) is not properly justified in the paper. Why do you use it instead of DBH? Saying we are dealing with young stands should not be enough. Is there any scientific reason to use do over DBH?

Thank you for the comment. We added the explanation in the text. We used d0 instead of DBH because some trees had a height under 130 cm, as can be seen in Figure 2, and Table 3. Therefore, DBH could not be measured for all trees and implemented in the models. Diameter d0 is perhaps not typical for purposes in forestry research but it is commonly used in biology, ecology etc.    

Some of the results are weak and others do not have the correct explanation:

The equation relating do and tree height has no statistical diagnosis except the R2. Authors should show this diagnosis (bias, RMSE, basic statistical problems...) if they are suggesting other authors to apply in order to use tree height as independent variable in biomass alometric equations. Error additivity should be taken into account.

Thanks you for the comment. We added additional information about the derived model between d0 and height.

Basic field data tables are missing. Even if some of the ranges are showed in the text, a table with max, min, mean and number of observations for each variable would help understand the results.

We added four tables with the description of site and stand conditions, sampled trees and foliage, measured plots.

Table 1: R2 are very poor. Even if parameters are significant, a broader statistical analysis is needed. Models with R2 of 0.09 to 0.34 do not explain at all the variability that may occur. Even accepting some of the less poor R2s, more robust statistical analysis is needed (residues homocedasticity, normality, VIF, RMSE...etc.)

Thank you for the comment. It is true that R2 coefficients are low in the case of leaf traits , but R2 of biomass models are high (Tables 7 and 9). Based on the comment of reviewers 1 and 3 we added generalised linear mixed models for leaf traits, the results of which are presented in Table 6. These models explain more than 50% of the variability of leaf traits, since their R2 is greater than 50. All models in Table 5 (originally Table 1) are presented including the information about MSE, from which RMSE can be calculated. Variance Inflation Factor (VIF) is used to quantify multicolinearity between independent variables. Our models in Table 5 have only one independent variable (d0), so VIF is not applicable. We could present residues in a graphical form, however, the extent of the paper would be substantially lengthened due to a large number of derived models.  

Table 3-Table 4: Some parameters are not significant (or slightly significant) with big MSEs in some cases.

Thanks to this comment we found one editing mistake in MSE of LAR model, the correct value is 165.93 instead of 72762, which was in the table before. All models and their b2 coefficients in these two tables are significant. From six b1 coefficients presented in Tables 3 and 4, four are significant at 95% significance level, and two are not significant at this level, but the p value of one of them is 0.082, which means that it is significant at 92% level, and the last one at 71% level. It is possible to test other models, but our aim was to apply the same approach for all dependent variables (possibility for mutual comparisons of the models).

In my opinion, overally bigger effort is needed in modelling procedures in order to avoid the shown problems of the models. Besides, there is no explanation of low R2s or non-significance of some parameters. Nevertheless, more modelling effort is needed.

Thank you for the comment. We built additional generalized linear mixed models to express the influence of tree diameter and position in crown on foliage traits.

The discussion correctly shows the analysis of the results in terms of allocation, even if those could be shown as trends also in results.

Thank you for the comment. We added some text focusing on the trends in results section.

Some of the assumptions must be taken with care: The point of maturity the trees reach the maximum LAI should not be assumed as comparison with other species and locations. The lack of data for adult trees in the species is not enough reason to assume some comparisons.

Thank you for the comment. We deleted the assumption about the possible trend of LAI in aspen with age.

In general, I find the study interesting, with a good introduction, methodology design and a correct discussion. Nevertheless, all the modelling procedures should be improved, together with the way of exposing the results.

We performed additional modelling and derived a generalized linear mixed model for leaf traits. Moreover, we also changed interpretation of results in Results and Discussion sections to a large extent.  

Round 2

Reviewer 3 Report

Thank you for the extensive changes developed in the paper. Most of the ideas shown in the first review have been correctly addressed. Introduction has been clearly enhanced and the basic survey data have been included.

Nevertheless, the models regarding leaf traits are still not valid. The adding of the GLMM approximation is interesting and gives more soundness to the results. But models and table 5 is mantained, and the statistical diagnosis is not acceptable. The fact that biomass equations have better statistical diagnosis has no effect on the leaf traits models. And these ones are not acceptable in terms of R2 or MSE, for instance. Those models can not be used to extract any conclusion due to a lack of statistical soundness. Once the GLMM approach is done and has good results, authors should rethink on the use of the previous models (analyzed in Table 5) because they are not useful.

Tables 8 and 9 (previous tables 3 and 4): The fact that some of the parameters are significant does not allow other parameters to be non-significant. If authors accept significance at 92% and, especially at a 71%, they should be able to explain the reasons to accept this significance.

Finally, I mantain my doubts on the possible heterocedasticity of the models. Minimum text and/or graphic explanation should be helpful. 

Author Response

Responses to Editorial office and Reviewer No. 3

Query from the Editorial Office:

The authors could better explain why they decided to maintain the first model; its statistical validity and utility.

Based on the query from the Reviewer No 3. we decided to delete Table 5, which showed the statistical characteristics of the “old” model for leaves.              

Moreover, we fulfilled another query from the e-mail of the Editor and added the Conclusions section in the end of the main text.

Query from the Reviewer No. 3

Nevertheless, the models regarding leaf traits are still not valid. The adding of the GLMM approximation is interesting and gives more soundness to the results. But models and table 5 is maintained, and the statistical diagnosis is not acceptable. The fact that biomass equations have better statistical diagnosis has no effect on the leaf traits models. And these ones are not acceptable in terms of R2 or MSE, for instance. Those models can not be used to extract any conclusion due to a lack of statistical soundness. Once the GLMM approach is done and has good results, authors should rethink on the use of the previous models (analyzed in Table 5) because they are not useful.

 We agree with the opinion of the reviewer (and also with the comment from the Editorial Office) and we deleted Table 5 with old models of leaf traits and related text. Consequently, the attention is focused on the new (mixed) model for foliage traits (actual Table 5).  

Tables 8 and 9 (previous tables 3 and 4): The fact that some of the parameters are significant does not allow other parameters to be non-significant. If authors accept significance at 92% and, especially at a 71%, they should be able to explain the reasons to accept this significance.

 Thank you for the comment. We tried models after exclusion of insignificant parameters b1, i.e. y=x**b2 instead of y=b1*x**b2 but we found out that R2 of these models was substantially reduced, e.g. the model for woody parts stock was reduced from 0.882 to 0.507. This shows that although the coefficients are not significant at α=0.05, they improve the model performance, and therefore, it is better to leave them in the model than to exclude them.

Finally, I maintain my doubts on the possible heteroscedasticity of the models. Minimum text and/or graphic explanation should be helpful. 

 A logarithmic version of the equation is frequently used to exclude heteroscedasticity of residues, which is always present in the calculation of biomass components. Using logarithmic equation enables the estimation of parameters with linear regressions, which fulfils the assumption of constant variance of residues. In the last years, non-linear regression methods have been developing based on which the question has arisen if it is not better to use the allometric equation in the power form and thus to exclude the logarithmic transformation. “Linearisation” allows using known methods of regression analysis and simpler calculation, especially if more independent variables are included. However, a disadvantage of this approach is that logarithmic transformation deforms original data due to which a correction factor should be used for its back transformation. The opinions on these two methodological approaches differ between papers, e.g. Cienciala et al. (2006), Lai et al. (2013), Mascaro et al. (2013). Moreover, some of our previous papers (Pajtík et al. 2018) showed that linear regression may lead to errors in determining biomass component.

Regarding the comment from the reviewer, we added extra paragraph to the Material and methods (Calculations and modelling) section to explain briefly about different methodological approaches related to heteroscedasticity of data.